# Modeling the t(2;5) Translocation of Anaplastic Large Cell Lymphoma Using CRISPR-Mediated Chromosomal Engineering

**DOI:** 10.3390/cancers17132226

**Published:** 2025-07-02

**Authors:** Robin Khan, Laurent Phely, Sophia Ehrenfeld, Tatjana Schmitz, Pia Veratti, Jakob Wolfes, Khalid Shoumariyeh, Geoffroy Andrieux, Uta S. Martens, Stephan de Bra, Martina Auer, Oliver Schilling, Melanie Boerries, Michael Speicher, Anna L. Illert, Justus Duyster, Cornelius Miething

**Affiliations:** 1Department of Medicine I, University Medical Center Freiburg, University of Freiburg, 79106 Freiburg, Germany; robin.khan@hotmail.de (R.K.); tatjana.schmitz@uniklinik-freiburg.de (T.S.); pia.veratti@uniklinik-freiburg.de (P.V.); khalid.shoumariyeh@uniklinik-freiburg.de (K.S.); stephan.bra@uniklinik-freiburg.de (S.d.B.);; 2Department of Pediatrics, University Hospital Würzburg, 97080 Würzburg, Germany; 3Department of Internal Medicine II, Hematology, Oncology, Clinical Immunology and Rheumatology, University Hospital Tübingen, 72076 Tübingen, Germany; 4Institute of Molecular Medicine and Cell Research, University of Freiburg, 79106 Freiburg, Germany; 5German Cancer Consortium (DKTK), Partner Site Freiburg, 79106 Freiburg, Germany; melanie.boerries@uniklinik-freiburg.de; 6Institute of Medical Bioinformatics and Systems Medicine, Medical Center—University of Freiburg, Faculty of Medicine, University of Freiburg, 79106 Freiburg, Germany; geoffroy.andrieux@uniklinik-freiburg.de; 7Institute of Human Genetics, Medical University of Graz, 8036 Graz, Austria; martina.auer@medunigraz.at (M.A.);; 8Institute of Surgical Pathology, University Medical Center Freiburg, University of Freiburg, 79106 Freiburg, Germany; 9Department of Medicine III, Faculty of Medicine, Klinikum Rechts der Isar, Technical University Munich (TUM), 80333 Munich, Germany; 10Center of Personalized Medicine, Technical University of Munich (TUM), University Hospital, Technical University of Munich, 80333 Munich, Germany

**Keywords:** ALCL, Npm-Alk, CRISPR/Cas, oncogenic fusion protein, chromosomal translocation, chromosomal engineering

## Abstract

ALK+ anaplastic large cell lymphoma (ALCL) is an aggressive lymphoma characterized by the presence of the nucleophosmin-anaplastic lymphoma kinase (*NPM-ALK*) oncogene resulting from a chromosomal rearrangement between chromosome 2 and chromosome 5 which drives lymphomagenesis. To improve current ALCL research models using artificial overexpression of *NPM-ALK*, we developed a CRISPR/Cas-based model which selectively introduces syntenic *Npm-Alk* translocations in a murine model cell line, leading to faithful Npm-Alk expression from the endogenous promoter and a more accurate recapitulation of the disease phenotype.

## 1. Introduction

Anaplastic large cell lymphoma (ALCL) is a rare disease accounting for 10–15% of the pediatric and 2% of adult non-Hodgkin lymphomas (NHLs), typically harboring the translocation t(2;5)(p23;q35) joining the tyrosine kinase anaplastic lymphoma kinase domain (*ALK*) to the nucleophosmin domain (*NPM*) located on chromosome 5q35 [1]. The latest classification in the revised fifth edition of the WHO in 2022 includes entities of ALCL depending on the presence of *ALK* translocations, the so-called *ALK*-positive (ALK+) and *ALK*-negative (ALK−) ALCL and the breast implant-associated ALCL (BIA-ALCL) [2]. Most children and young adults show an *ALK*+ status, while older patients tend to be *ALK*- [3]. *ALK*- ALCL patients lean towards a worse outcome with a 5 year median overall survival of 48% compared to 80% in patients with *ALK*+ disease [4,5]. 

The t(2;5) chromosomal translocation represents the defining initial event in *ALK*+ ALCL lymphomagenesis. The formation of interchromosomal translocations requires a double-strand break (DSB) on each of the chromosomes simultaneously and the subsequent inaccurate joining of the DNA ends [6], in which a type of non-homologous end joining (NHEJ) is typically involved. Thereby, a fusion gene is generated by juxtaposing two normally separate genes. The new position either leads to overexpression of genes with oncogenic properties through transfer to a strong promoter, or expression of a novel fusion protein that exhibits oncogenic potential.

Like other tyrosine kinases, ALK activates through homo-dimerization upon ligand binding and is deactivated through dephosphorylation in the absence of the ligand [7]. The ALK protein is involved in activating several signaling pathways such as the phospholipase C-γ (PLC-γ), signal transducer and activator of transcription 3 (STAT3) and phosphatidylinositol-3-kinase (PI3K) pathways [8,9].

The *NPM* gene encodes for three transcript variants which share significant sequence and structural homology [10]. The nucleolar multifunctional phosphoprotein (Nucleophosmin) isoforms (35–40 kDa) are expressed in all tissues and have abundant functions including metabolic pathways for mRNA processing, chromatin remodeling and embryogenesis as well as pathways in DNA repair and regulating apoptosis [11]. The role of *NPM* within the t(2;5) translocation is twofold: First, the strong *NPM1* promoter leads to the aberrant overexpression of the ALK tyrosine kinase in lymphoid cells. Furthermore, the N-terminal NPM fragment leads to the multimerization of ALK, inducing autophosphorylation and dysregulated activation [12], thus promoting excessive initiation of signaling cascades responsible for cell growth, transformation and anti-apoptotic processes [13]. In recent years, several studies have demonstrated the use of gene-editing systems, such as transcription activator-like effector nucleases (TALEN) and CRISPR-Cas9, to induce specific chromosomal translocations and intrachromosomal inversions in vitro, including MLL-AF9 [14,15], BCR-ABL1 [16], RUNX1-RUNX1T1 [17] and others [18,19].

Building on this work, we aimed to create a CRISPR/Cas9-based model leading to the expression of *Npm-Alk*, the major driving oncogene in *ALK*+ ALCL. For this purpose, we established a retroviral delivery system for Cas9 and two distinct sgRNAs targeting the *Npm1* and *Alk* gene locus in murine cells. With this method, we were able to generate an improved model of ALCL recapitulating the genetic events occurring during human oncogenesis. 

In contrast to previous approaches, including the use of retroviral vectors combined with the Cre-Lox system to conditionally express *Npm-Alk* from heterologous promoters [20], this method directly replicates the t(2;5) chromosomal translocation. Furthermore, our workflow can potentially be expanded to model other relevant chromosomal translocations in vitro and in vivo.

## 2. Materials and Methods

### 2.1. Retroviral Expression Constructs and Cloning of the GBAC Vector System

The MSCV-MCS-PGK-Puro-IRES-GFP plasmid was a gift from Christopher Vakoc (Addgene plasmid # 75,124) [21]. The MSCV_Cas9_Puro plasmid was a gift from Christopher Vakoc (Addgene plasmid # 65,655) [22]. The eSpCas9 (1.1) (enhanced Streptococcus pyogenes Cas9) plasmid was a gift from Feng Zhang (Addgene plasmid # 71,814) [23]. The pX330-U6-Chimeric_BB-CBh-hSpCas9 plasmid was a gift from Feng Zhang (Addgene plasmid # 42,230) [24].

### 2.2. Golden Gate Cloning

To facilitate cloning of individual gRNAs as well as larger gRNA libraries, we developed a recipient vector platform enabling Golden Gate cloning of gRNAs [25]. The recipient vector included a CcdB cassette flanked by type IIs restriction enzyme sites to reduce empty background clones. In this study, Golden Gate cloning was performed alternatively to the restriction digest and DNA ligation to clone single guide RNAs (sgRNAs) into the GBAC vector in a one-step approach. First, the gRNA was synthesized as single-strand DNA (Merck, Darmstadt, Germany), which was then amplified by PCR using primers containing compatible type IIs restriction sites to generate double-stranded templates for Golden Gate cloning. Subsequently, the template and recipient vector were mixed in a ratio of 3:1, and 1 µL of the Type IIS restriction enzyme BspQI (New England Biolabs, Ipswich, MA, USA) and 1 µL of T4 DNA ligase (New England Biolabs, Ipswich, MA, USA) was added. By using 10 cycles of heating up to 37 °C for 2 min and cooling down to 16 °C for 5 min the optimum temperature range for the activity of BspQI and DNA ligase was utilized, respectively. After completion of the Golden Gate procedure, the ligation reactions were used for transformation of competent *E. coli* bacteria.

### 2.3. Surveyor Nuclease Assay

This assay was used to detect DNA mutations like small insertions or deletion, as well as single base mismatches. The surveyor endonuclease cleaves all heteroduplexes generated by hybridizing DNA fragments. Afterwards, analysis was performed by agarose gel electrophoresis. At first, PCR amplification at the region of interest was performed using specifically designed primers. Efficiency of the PCR was monitored using 5 µL of each product in agarose gel electrophoresis. The remaining 45 µL were treated as instructed by the mutation-detection kit (Surveyor^®^, IDT^®^, Coralville, IA, USA).

### 2.4. Cut-Point-Analysis by Deep Sequencing

To prepare the genomic DNA for deep sequencing, the target sequences were amplified via PCR using specifically designed primers containing the required p7 and p5 overhangs for Illumina sequencing. A 6-base barcode is embedded into the p7 primer to allow custom multiplexing of PCR products, the p5 primer contains the binding site for the sequencing primer.

To balance the amounts of reads between each sample for a deep sequencing run, we performed a gel electrophoresis of the PCR products and determined the required DNA amount to load individually. The determined DNA amounts are mixed together, column-purified and measured using the Qubit^®^ dsDNA HS Assay kit (Invitrogen, Waltham, MA, USA). The displayed DNA concentration is then diluted to 80 ng/µL. MiSeq or NextSeq 500 tabletop sequencing machines (Illumina, San Diego, CA, USA) were used for deep sequencing of the cut-point PCR products using SR cartridges according to manufacturer’s protocol. After demultiplexing, the reads are aligned to the target sequence and the cut rate is determined by calculating the ratio between mutated/modified to wildtype sequences.

### 2.5. Multiplex Fluorescence In Situ Hybridization (M-FISH)

This technique allows studying of interchromosomal rearrangements by tagging homologous chromosome pairs with a finite number of fluorophores. The genome is then visualized by microscope and the data are acquired digitally. This method was performed using the Metasystems’ Multi-color Probe Kit (MetaSystems, Altlussheim, Germany) according to manufacturer’s protocol, stained cells were analyzed using the *Cytovision software* (Leica Biosystems, Wetzlar, Germany; https://www.medicalexpo.com/prod/leica-biosystems/product-95735-653665.html, accessed on 25 June 2025).

### 2.6. Western Blotting and Immunodetection

Protein lysates were analyzed using standard Western blot protocols. Briefly, protein lysates were separated by SDS page gel electrophoresis and immobilized on a PVDF membrane (Immobilon^®^, Merck Millipore, Burlington, MA, USA) by western blotting. The membrane was incubated in blocking buffer (Odyssey^®^, Licor, Lincoln, NE, USA) prior to incubation with primary antibody diluted in blocking buffer overnight at 4 °C on a shaker. After several washing steps and incubation with secondary antibodies diluted in blocking buffer, the blot was visualized using an IR Imaging System (Odyssey^®^, Licor, Lincoln, NE, USA).

### 2.7. Data Analysis and Presentation

For the statistical analysis, GraphPad Prism software was used (GraphPad Software, Inc., version 9.4.0, Boston, MA, USA). As we compared a minimum of 4 groups over several timepoints, the two-factorial analysis of variance (two-way ANOVA) with the Bonferroni post-hoc test was applied. Significance was stated, whenever the probability value (*p*-value) was *p* < 0.05 (*), *p* < 0.01 (**) or *p* < 0.001 (***). Gene enrichment levels were assessed with help of the GSEA software (Broad Institute, Cambridge, MA, USA; UC San Diego, San Diego, CA, USA, available at: https://www.gsea-msigdb.org, accessed on 27 June 2025) [26,27,28].

### 2.8. Culturing of Cell Lines

Cells were cultured under humidified conditions at 37 °C and 5% CO_2_ atmosphere using standard cell culture techniques. Ba/F3 (Cat# ACC 300, DSMZ, Braunschweig, Germany), HL60 (Cat# ACC 3) and K562 (Cat# ACC 10) cells were grown in RPMI supplemented with 10% FCS, 1% Pen-Strep and 1% Glutamine, as well as recombinant IL-3 (2 ng/mL, PeproTech, Hamburg, Germany) in the case of Ba/F3 cells. In selected cases, Ba/F3 cells were grown without IL-3 to select for *Npm-Alk*-mediated transformation. Murine MCL22-1 primary lymphoma cells were cultured on irradiated B6 mouse embryonic fibroblasts (MEF) or irradiated Arf/p19 knockout MEFs that serve as feeder cells to support cell growth, using BCM (50% IMDM/50% DMEM) media supplemented with 10% FCS, 1% Pen/Strep (100 U/mL; 100 μg/mL) and 50μM ß-mercaptoethanol. Plat-E cells were grown in DMEM with 10% FCS, 1% Pen-Strep and 1% Glutamine according to standard protocols [29].

### 2.9. IL-3 Deprivation of Ba/F3 Cells

Cells with *Npm-Alk* translocation events gained the expression of an oncogene. This oncogene substituted the IL-3 signaling in Ba/F3 cells. Selection of these cells was performed by depriving the cell culture medium of IL-3. 3d after third infection, Ba/F3 cells were centrifuged at 1500 rpm for 5 min and washed with PBS once to remove all IL-3 containing growth media. Afterwards, the cells were resuspended in RPMI (10% FCS, 1% P/S). Viability of cells was monitored every 2 d via flow cytometry. Once IL-3 independent cells grew to 500,000/mL, they were split ¼ every 2 days.

### 2.10. Retrovirus Production and Infection

To produce recombinant retroviral particles, Plat-E cells (Cell Biolabs, San Diego, CA, USA) were transfected with retroviral expression plasmids using transfection reagents according to manufacturer’s protocol (TurboFect^®^, Thermo Fisher Scientific, Waltham, MA, USA). Supernatant of transfected Plat-E cells was collected, filtered and added to the target cells with the addition of 4 µg/mL polybrene before spin infection by centrifuging for 15 min at 1500 rpm. This step was repeated 12 and 24 h later after removing the supernatant. Infection rates were monitored via flow cytometry of GFP expression 24–48 h after the last infection step.

### 2.11. Flow Cytometry

A cell analyzer (BD LSRFortessa™, BD Biosciences, San Jose, CA, USA) was used to monitor viability (FSC, SSC) and expression of fluorescent proteins. A minimum of 5000 cells was analyzed in each measurement. Flow cytometric data were analyzed using the FlowJo software, version 10 (BD Biosciences, Ashland, OR, USA).

### 2.12. Cell Sorting

Retrovirally infected cells were sorted by fluorescence-activated cell sorting (FACS) according to the expression of fluorescent proteins. This method was performed to select cells without using antibiotics by filtering them according to their expression of green fluorescent protein (GFP). Cell sorting was performed on a BD FACSAria™ III Cell Sorter (BD Biosciences, San Jose, CA, USA) by our Core Facility, Department of Medicine I, University Medical Center Freiburg.

### 2.13. Gene Expression Analysis

For gene expression analysis of Ba/F3 cells transformed by CRISPR-mediated *Npm-Alk* translocation, transformed by *Npm-Alk* overexpression or control Ba/F3 cells, RNA was isolated from cells growing in steady-state using Trizol (Thermo Fisher, Waltham, MA, USA). Gene expression was analyzed using the Affymetrix GeneChip^®^ Gene 1.0 ST Array System (Thermo Fisher). Probe generation, hybridization and scanning was performed according to the manufacturers’ protocol. All experiments were done in triplicate.

## 3. Results

### 3.1. Establishment of a CRISPR/Cas9-Based Vector System

For the generation of a CRISPR/Cas9-based system inducing *Npm-Alk* translocations, we developed a retroviral gRNA expression system (GBAC—gRNA-pgk-blasticidin-2A-mCherry) based on a self-inactivating (SIN) vector backbone (pQCXIX, Clontech), enabling the expression of gRNAs from a U6 promoter cassette together with expression of a blasticidin resistance gene for eukaryotic selection and the mCherry fluorescent protein to facilitate detection of infected cells by FACS (Figure 1a). For double-infection experiments using two different gRNAs, we exchanged the mCherry fluorescent protein with the cyan fluorescent protein gene generating the GBACy vector. In order to enable easy shuttling of gRNAs into the GBAC vector by Golden Gate cloning [30], we included a ccdB cassette flanked by BspQI Type IIs restriction enzyme sites downstream of the U6 promoter. For simultaneous expression of two gRNAs from the same vector, we devised a strategy to clone 2 U6 promoter-gRNA cassettes in tandem, enabling dual targeting of two distinct sites with one vector (see also Section 2).

For Cas9 or espCas9 expression, we utilized the tetracycline-inducible T3GCasVIN vector, which is tagged with yellow fluorescent protein and contains a neomycin resistance gene, as well as the constitutively active MIG_Cas9 and MIG_eCas9 variants, which co-express green fluorescent protein and confer puromycin resistance (Figure 1b).

### 3.2. Functional Validation of Cas9 Expression Vectors After Retroviral Infection of Target Cells

To test the efficiency of the novel retroviral expression vectors, Ba/F3 cells were infected with Cas9 expression vectors and tested for the functionality of Cas9. The Ba/F3 cell line is a murine pro-B cell line that is strictly dependent on interleukin-3 (IL-3) for survival and proliferation. Introduction of oncogenic drivers, such as *Npm-Alk*, into Ba/F3 cells confers IL-3 independence, which serves as a robust readout for cellular transformation [31]. We furthermore included the human leukemia cell lines K562 and HL60, as well as a primary murine lymphoma cell line in the initial analysis.

During testing of the tet-inducible Cas9 expression constructs, Ba/F3 and K562 cells showed strong Cas9 expression levels under doxycycline (dox) treatment (+) with low but residual Cas9 expression levels in the absence of doxycycline (−) (Figure 1c, upper panel). Conversely, no relevant Cas9 expression could be shown in HL-60 and MCL 22-1 samples irrespective of dox treatment. After infection with MSCV-Cas9_PIG (C_PIG) or eCas9_PIG (eC_PIG) and short selection using puromycin, the Western blot analysis showed that infected Ba/F3 cells expressed Cas9, whereas non-infected Ba/F3 cells showed no expression as expected (Ø) (Figure 1c, lower panel). Notably, the expression of Cas9 in Ba/F3_C_P was consistently slightly weaker compared to Ba/F3_eC_PIG.

**Figure 1 cancers-17-02226-f001:**
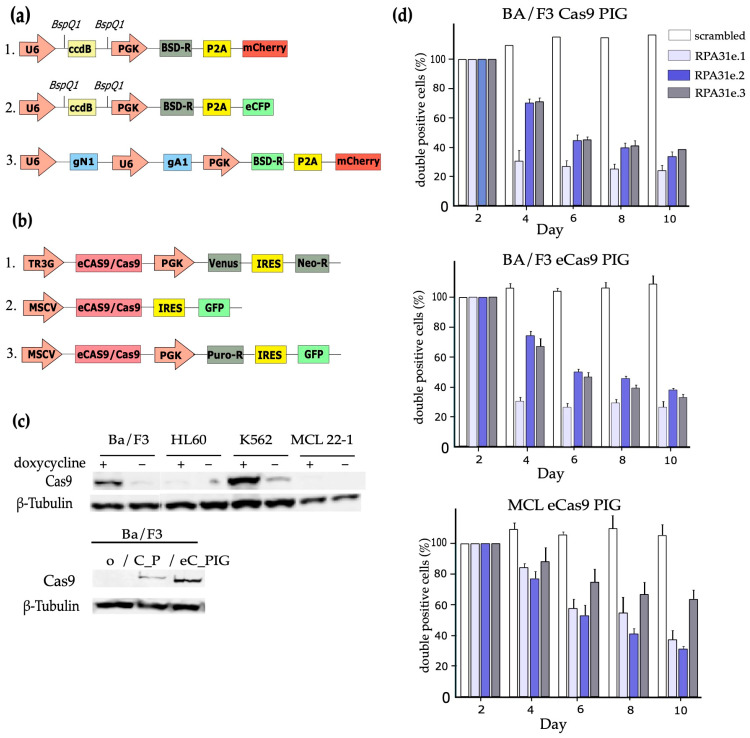
(**a**) sgRNA expression vector outline: 1. GBAC_Rec: Constitutive sgRNA expression with sgRNA landing pad containing a ccdB cassette to facilitate Golden Gate cloning and mCherry as marker (size: ~7.2 kb). 2. GBACy_Rec: Constitutive sgRNA expression with sgRNA landing pad containing a ccdB cassette to facilitate Golden Gate cloning and CFP (cyan fluorescent protein) as marker (size: ~7.2 kb). 3. 2GBAC_Npm1-Alk: dual sgRNA expression vector enabling cistronic expression of two gRNAs (in this case sgRNAs targeting *Npm1* and *Alk*): ~7.6 kb. (BSD-R: blasticidine resistance gene; BspQI: BspQI recognition site; eCFP: enhanced cyan fluorescent protein; gA1: sgRNA targeting *Alk* (Alk_1); gN1: sgRNA targeting *Npm*1 (Npm1_1); mCherry: fluorescent protein; P2A: peptide 2A sequence; PGK: phosphoglycerate kinase; U6: Human U6 promoter). (**b**) Cas9 expression vector outline: 1. T3GCasVIN: vector for tetracycline inducible Cas9 expression (size: ~12.4 kb). 2. MIG_Cas9/MIG_eCas9: Two constitutive variants, one with Cas9 and one with eCas9, coupled to GFP expression. (size: ~10.7 kb.) 3. Cas9_PIG/eCas9_PIG: Two constitutive variants, one with Cas9 and one with eCas9, constitutive expression puromycin, (size: ~12 kb.) (Cas9: CRISRP-associated protein9; eCas9: “enhanced” Cas9; GFP: green fluorescent protein; IRES: internal ribosome entry site; MSCV: murine stem cell virus promoter; Neo-R: neomycin resistance gene; PGK: phosphoglycerate kinase; Puro-R: Puromycin resistance gene; TR3G: tetracycline response element 3rd generation; Venus: yellow fluorescent protein). (**c**) Cas9 protein expression in different murine and human cell lines using inducible and constitutive expression vectors. Western Blot (WB) analysis of Cas9 expression in Ba/F3, HL-60, K562 and MCL22-1 cells infected with rtTA3-IRES-EcoR-puromycin vector (RIEP) and the T3GCasVIN construct, selection with puromycin/neomycin and 2d of doxycycline treatment (upper panel). Cas9 expression in Ba/F3 using constitutive Cas9 expression vectors (Cas9_PIG/eCas9_PIG). Original western blots are presented in Appendix A. (**d**) Flow cytometric monitoring of cells after Cas9-induced targeting of the essential gene RPA3 using 3 different sgRNAs. Over 10 days in culture, the cells carrying a gRNA directed against RPA3 depleted significantly compared to control (*p*-value of at least *p* < 0.01) in Ba/F3 cells transduced with Cas9 or eCas9 constructs, as well as in the primary murine lymphoma line MCL22-1 expressing eCas9. For further functional testing of our CRISPR/Cas9 system, we targeted Rpa3, a known essential gene in most dividing eukaryotic cells [32] using three different gRNAs from a previous publication [22]. Thereby, Cas9-induced Rpa3 inactivation should lead to depletion of cells expressing those sgRNAs compared to control cells. As a negative control we used a scrambled sgRNA that showed no exon target in BLAST (NCBI Primer-BLAST, available at: https://www.ncbi.nlm.nih.gov/tools/primer-blast/, accessed on 27 June 2025) [33]. For initial testing purposes, we used Ba/F3 cells and a murine Mantle Cell Lymphoma (MCL) line expressing either Cas9 or eCas9 after bulk infection and selection, and subsequently infected these cells with the different Rpa3 or scrambled control gRNAs. Using flow cytometry to measure the GFP+/mCherry+ cell fraction over time, selective enrichment or depletion of gRNA expressing cells under normal growth conditions was analyzed. The relative abundance of infected cells carrying a gRNA targeting Rpa3 rapidly depleted over time (Figure 1d), suggesting efficient Cas9-mediated inactivation of Rpa3 in this model.

While we did not observe significant differences in depletion between Cas9- and eCas9-expressing Ba/F3 cells, we noticed significant depletion of the Cas9, and to a smaller extent of the eCas9 construct in the murine MCL line even prior to gRNA coexpression, suggesting increased toxicity of the original Cas9 construct in these cells.

The increased toxicity observed with the original Cas9 construct in certain cells has been described by other groups as well, and seems to be associated with its activation of a p53-mediated DNA damage response (DDR) [34,35,36]. When CRISPR-Cas9 induces double-strand breaks (DSBs), cells with functional p53 pathways respond by initiating cell cycle arrest or apoptosis to mitigate genomic instability. Accordingly, we observed significant toxicity of Cas9 expression in primary cells (murine BM cells) or primary cell lines (e.g., the primary murine lymphoma line Mcl 22-1), which may retain more elements of their DDR including the p53 pathway compared to established cell lines. 

We therefore preferentially used the eCas9 construct in the following experiments.

### 3.3. CRISPR/Cas9 Induced Npm1-Alk Translocation

To induce the generation of *Npm-Alk* translocations in live cells, we applied CRISPR-Cas9-mediated double-strand breaks at defined syntenic regions in the mouse genome corresponding to the predominant intronic breakpoint sites in human ALCL. A schematic overview of the approach used to generate the *Npm-Alk* translocation in murine cells is provided under Figure 2a, a detailed map of the corresponding chromosomal regions and the intron/exon structure of Npm1 and Alk highlighting the major breakpoint sites are shown in Appendix A.

We designed three separate sgRNAs targeting predetermined breakpoint regions in *Npm*1 (gN1, gN2, gN3) and *Alk* (gA1, gA2, gA3) (Appendix A) and cloned them into the fluorescently tagged GBAC and GBACy delivery vectors, allowing us to monitor sgRNA expression by flow cytometry. All experiments were performed in duplicates.

MSCV_Cas9_Puro Ba/F3 cells were separately infected with each sgRNA combination on day 0. After confirming successful infection via flow cytometry, the cells were deprived from IL-3 and monitored over the course of 14 days.

After an initial rapid drop in viability, the infected cells expressing sgN1 and sgA1 outgrew the uninfected control cells and other gRNA combinations in the absence of IL-3, starting on day 6 (Figure 2b). Similar results could be observed by swapping the Cas9 expressing MSCV_Cas9_Puro vector with the C_PIC or eC_PIC vector or by exchanging the sgRNA delivery vector to the 2GBAC vector, expressing sgN1 and sgA1 simultaneously from a single expression construct (Figure 2b). 

After 14 days in culture, the cells were harvested and lysed for western blotting and gDNA extraction. Non-infected Ba/F3 (ø) and Ba/F3 infected only with the Cas9-expressing vector (C_PIC) were used as negative controls. Western blot analysis showed a strong expression of a 75 kDA hybrid Npm-Alk fusion protein (Figure 2c) and their respective β-tubulin control.

By using a forward primer 250 bp upstream of the Alk cutting site and a reverse primer 250 bp downstream of the *Npm1* cutting site we were able to amplify a 554 bp long product using PCR (Figure 2d). By Sanger sequencing, all sequences showed on target cleavage approximately 3 bp upstream of the PAM-sequence resulting in non-identical double-strand breaks around the targeted breakpoints (Figure 2e).

Next, we used multiplex fluorescence in situ hybridization (M-FISH) to check for the presence of interchromosomal translocations. MSCV-Cas9_Puro cells infected with 2GBAC_Npm1-Alk or double infected with GBAC_Npm1_1 and GBACy_Alk_1 presented with a t(11;17) chromosomal translocation (Figure 2f, marked in red). Uninfected Ba/F3 cells and Ba/F3 infected with a *Npm1-Alk* overexpressing (MSCV_NPM-ALK_IRES_eGFP) plasmid were used as control cell lines. Here, we did not observe the t(11;17) translocation, but the control cell line overexpressing *Npm-Alk* showed several different DNA alterations including derivative translocations der(3)t(3;6) and der(8)t(2;8) as well as several numerical chromosomal aberrations, indicating an ongoing genetic drift in these cells (Appendix A). 

In spite of testing three different combinations of gRNAs for translocation induction, only one specific gRNA combination was able to induce *Npm-Alk* chromosomal translocations leading to the outgrowth of transformed cells (Figure 2b). To dissect the factors influencing the efficacy of fusion gene generation, we investigated gRNA cleavage efficiency via deep sequencing. We isolated gDNA from Ba/F3-Cas9 cells retrovirally infected with a single gRNA (gA1, gA2, gA3, gN1, gN2, gN3) and amplified the cutpoint region using locus-specific primers, which also incorporated a barcode sequence for multiplexing (see Section 2). By high-throughput sequencing of more than 10^5^ reads per locus, we were able to quantify the number of Cas9-modified sequences compared to wildtype for each gRNA. The deep sequencing analysis showed that gN1 and gA1 led to an on-target mutation rate of 70% and 90%, respectively, while the remaining gRNAs showed a cleavage efficiency of below 45% (Figure 2g). Our approach, while probably still underestimating the total number of effectively modified cells due to missing large deletions encompassing the primer binding site, indicates that highly efficient gRNAs are required to induce functionally active translocations in this setting.

### 3.4. CRISPR-Cas9-Induced Npm-Alk Fusions Respond to Alk-Inhibition and Show a Distinct mRNA Expression Pattern

The results of the M-FISH corroborated the Western blot analysis data demonstrating that the established CRISPR/Cas9 system induced *Npm-Alk* translocations in Ba/F3 cells. The efficacy of *Npm-Alk* as an oncogene driving the outgrowth of Ba/F3 cells was further tested using the specific Alk inhibitor NVP-TAE684 [37] (Figure 3a). The inhibitor was added to the cell culture medium for two days and the cell viability was monitored via flow cytometry.

Two Ba/F3 cell lines growing IL-3 independent after translocation induction (Ba/F3 infected with MSCV_Cas9_Puro and GBAC_Npm1 + GBACy_Alk_1), and Ba/F3 transduced with MSCV_Cas9_Puro_2GBAC_Npm1-Alk1) were treated with different concentrations (5 nM, 10 nM, 50 nM and 100 nM) of NVP-TAE648. Additionally, Ba/F3_MSCV_Cas9_Puro growing in IL-3 supplemented media were used as control cells. NVP-TAE684 had no significant effect at any utilized concentration on the control cells growing without the t(11;17) translocation. The viability of the cells carrying the *Npm-Alk* translocation, however, decreased already 24 h after treatment with NVP-TAE684 (Figure 3a). The decrease in cell viability correlated with increased inhibitor concentrations, cells treated with the lowest concentration of NVP-TAE684 at 5 nM showed a slight decrease of cell viability of around 5–7% compared to untreated cells on day 1, whereas viability in *Npm-Alk* translocated cells treated with 50 nM and 100 nM decreased to <2% after two days for all samples analyzed.

### 3.5. Cells Carrying an Endogenous Npm-Alk Translocation Demonstrate a Different mRNA Expression Pattern than Ba/F3 Cells Transformed by the Overexpressed Fusion Protein

Although both cells exogenously overexpressing the Npm-Alk fusion protein from a heterologous promoter and cells expressing the fusion protein from the endogenous Npm1 promoter grow IL-3 independent with similar growth kinetics, there are several differences between the two approaches. The endogenous Npm1 promoter responds to different cues than the retroviral MSCV promoter used for overexpression of the fusion oncoprotein. Furthermore, the translocation event may also impact other genes positioned in proximity to the breakpoint locus, which may further influence the biological behavior of the tumor cell.

To identify more subtle differences between Ba/F3 cells transformed by either retroviral Npm-Alk overexpression or by endogenous CRISPR-mediated translocation, we performed mRNA expression analysis using Affymetrix GeneChip® Gene 1.0 ST Array System (Affymetrix, Santa Clara, CA, USA), to compare mRNA expression in steady-state growing Ba/F3 cells transformed by either MSCV-NPM-ALK or CRISPR-mediated recombination, with untransformed Ba/F3 cells growing in the presence of IL-3 as control. A principle component analysis showed that the two Npm-Alk transformed Ba/F3 cell lines, while more related than the untransformed control cells, clustered in distinct groups (Appendix A), indicating that the different approaches of Npm-Alk expression also induced diverging gene expression. Accordingly, more than 800 genes were differently expressed between the two conditions with at least 1 logFC and adjusted *p*-value <0.05 (Figure 3b). A heatmap of the top-differentially expressed genes in the three conditions did not show obvious differences in active signaling pathways (Figure 3c and Appendix A). To further dissect the pathways underlying the differential gene expression, we performed gene set enrichment analysis (GSEA) on the expression data from the three cell lines (Figure 3d). Interestingly, the GSEA analysis highlighted the Unfolded Protein Response (UPR) pathway as one of the most significantly diverging pathways between the two Npm-Alk transformation approaches in several canonical GSEA pathways including the Reactome and WikiPathway subset (Figure 3d) We also performed GSEA to identify topological effects of the chromosomal structural changes in the CRISPR model on gene expression e.g., due to loss (or gain) of long range repressors or enhancers as has been described previously in other cases [39]. Notably, a GSEA analysis focusing on genes located on the involved chromosomal regions (chromosome 11 and 17 in our murine model) showed generally reduced expression of genes in proximity to the translocation locus (Appendix A), suggesting that expression of the more proximal located genes in our model is mostly repressed after induced translocation.

## 4. Discussion

Chromosomal translocations are among the most frequent genetic events in cancer. Chromosomal translocations frequently lead to gene fusions, with deregulated expression of activated kinases strongly promoting the growth and survival of cancer cells. To study the effects of fusion neo-genes on cell growth and survival, cDNA overexpression of hallmark lesions like the *Bcr-Abl* fusion oncogene in CML models have been used to decipher the intracellular signaling driving cellular transformation. While the exogenous overexpression of the cDNA of fusion oncogenes recapitulates important aspects of the oncogenic transformation process including the relevant signaling pathways, it does not consider the chromosomal changes underlying the translocation event. Therefore, the advent of CRISPR technology to engineer chromosomal breaks, inducing chromosomal translocations to recapitulate specific oncogenic events in cancer promises to further improve genetic cancer models. Since translocations involving the ALK kinase are recurrent driver events in a range of human cancers, we initially focused on the development of models based on the *NPM-ALK* t(2:5) translocation. To enable fast and efficient recombination, we established a toolbox for the retroviral infection of murine cells with Cas9 and sgRNAs targeting the genes *Npm1* and *Alk*, in which infection efficacy could be monitored through flow cytometric analysis of linked fluorescent markers. To test for successful on-target operation of our Cas9/gRNA approach, we showed expression of the Npm-Alk fusion protein in Western blot analysis, validated recombination by breakpoint sequencing, and performed direct visualization of the chromosomal rearrangements using M-FISH. Furthermore, Alk dependent growth after successful chromosomal rearrangement in IL-3-independent Ba/F3 cells was shown by demonstrating impaired cell viability upon Alk inhibitor treatment.

To establish the CRISPR-based translocation model, we adapted a retroviral self-inactivating (SIN) vector to enable stable gRNA expression and at the same time identification of infected cells by fluorescence, as well as positive selection using blasticidin. To achieve better control over CRISPR activity, we choose to express the Cas9 gene from a separate retroviral vector. 

In our initial analysis of different inducible and constitutive Cas9 expression constructs, we observed a significant leakiness of the tet-inducible system, irrespective of the Cas9 variant (Cas9 or eCas9), in spite of our using a third-generation tet-on [40]. This is shown in Figure 1d, where residual expression can be detected in K562 and Ba/F3 cells. Of note, we did not observe any expression in the primary murine lymphoma cell line Mcl 22-1, likely due to toxic effects of Cas9 expression in this cell line. Consistently, the constitutive Cas9 expression system showed a higher efficiency, manifested by higher depletion rates when targeting essential genes such as Rpa3, likely due to counterselection and outgrowth of poor inducer cells in the tet-inducible system (Appendix A). We therefore decided to proceed using the constitutive Cas9 expression variants for further testing, since the combination of leakiness and toxicity limited the advantage of the tet-inducible system over the constitutive variants. 

For translocation induction, we designed several sgRNAs targeting the *Npm*1 and *Alk* locus located on murine chromosome 11 and 17, respectively. Also, to further maximize infection rates, we cloned a retroviral vector enabling simultaneous expression of two sgRNAs (2GBAC). While this numerically almost doubled the initial number of cells containing both sgRNAs, there was no significant latency in outgrowth between using the dual sgRNA vector and the double-infection approach, likely due to the rapid doubling time of the transformed cells. Of note, the 2GBAC infected cells displayed identical translocations regarding location and size as the single infected cells.

While the expression as single or double sgRNA did not significantly affect the efficiency of the system, the sgRNA design had a more profound impact. Initially, three different gRNAs for each locus were selected from predictions based on the CCTop algorithm [41]. The analysis of individual gRNA cutting efficacy by deep sequencing showed remarkable differences in cleavage efficiency between the different gRNAs, with the single productive combination (gRNA Npm1_1 and ALK_1) leading to mutations in about 70% and more than 90% at the respective target site. Compared to the next best gRNAs targeting the same gene, there was an observed difference of about 40% in cleavage efficiency. Our results substantiate the need of gRNA efficacy testing and indicate that only highly efficient sgRNAs with high cut rates will enable efficient recombination. The results suggest that there is a non-linear effect on translocation efficiency, with a threshold of gRNA cleavage efficacy which is needed to effectively enable chromosomal recombination. Alternatively, there may be a required minimal number of cells with effectively cleaved sites on both chromosomes to enable productive translocation. With a calculated cutting rate (based on the measured cleavage efficiency of 70% for sgRNA NPM1_1 and 90% for ALK_1) of 0.62 for the most efficient (and only productive) gRNA combination versus a cutting rate of 0.12 (30% and 40%) for the second best combination, an almost 6-fold difference may at least in part also account for the striking difference in the efficacy to generate productive translocations between the different gRNAs. Finally, while in our case the chosen gRNA cutting positions within the intron sequence on murine chromosome 11 and 17 varied by little more than 100 bp, the flanking nucleotide sequence as well as the epigenetic state of the individual sites may also play a role in the rate of productive translocations induced by a particular gRNA. Ongoing research into the determinants of effective translocation induction will likely lead to further improvements of the efficiency of the model.

Interestingly, while the CRISPR-induced transformation model showed many similarities to the Ba/F3 model based on *Npm-Alk* cDNA overexpression, we observed subtle differences in growth rates and response to Alk inhibition. To further examine the differences, we analyzed mRNA expression in the different cell lines using microarray expression analysis. The results revealed significant differences in gene expression between the cell lines, including a significant upregulation of unfolded protein response (UPR) genes in the *Npm-Alk* cDNA overexpressing cells, suggesting that exogenous overexpression of *Npm-Alk* may lead to increased artificial cellular stress levels. Additionally, a major difference between the two *Npm-Alk* models could derive from the impact on chromosome structure induced by the chromosomal translocation between chromosome 11 and 17 in our murine model corresponding to the syntenic chromosomes 2 and 5 in the t(2;5) translocation in human ALCL [1]. Notably, the analysis of gene expression changes within confined regions adjacent to the translocation site indicated a dampening effect of the rearrangement on the expression of genes close to the translocation breakpoint, in line with previous reports on gene regulation in other translocation events [42]. Thus, while the molecular mechanisms underlying these effects are not completely understood, our data corroborate that chromosomal translocations may affect gene expression beyond the genes immediately involved in the translocation event, implying that concise replication of the genomic environment will be crucial for the accurate modeling of tumors driven by translocation events.

## 5. Conclusions

In conclusion, we established an efficient and versatile system for the induction of genomic translocations based on a SIN gamma retrovirus backbone for Cas9 and sgRNA delivery. By targeting the syntenic murine *Npm* and *Alk* loci, we were able to generate a genetically faithful model recapturing the chromosomal alterations underlying the oncogenic events underlying the t(2;5) translocation in ALCL. Furthermore, our results indicate that models based on overexpression of the *Npm-Alk* cDNA by exogenous promoters differ from models driven by accurate replication of the chromosomal translocation, in part due to the impact of the structural changes on genes located in the proximity of the translocation breakpoint. We are currently testing if these results can be extended to other cell lines and in vivo models. So far, the rate of transformation in primary cells has been substantially lower using the same gRNAs. We observed rapid depletion of Cas9 expressing cells in murine hematopoietic stem cells as well as in primary murine lymphoma cells, indicating an increased susceptibility to Cas9-mediated cytotoxicity in primary cells. We also so far failed to transform murine stem cells with the 2GBAC construct, indicating that an increased sensitivity to DNA damage may affect the translocation efficiency in primary cells. Thus, blunting the DNA damage response e.g., by targeted deletion of the p53 gene could increase the efficacy of translocation induction and facilitate the generation of CRISPR-induced translocation cancer models [33]. Nevertheless, the described model may help to further elucidate the contributing genetic factors underlying the development of ALCL and pave the way for the development of other genetically faithful models of translocation-driven cancers.

## Figures and Tables

**Figure 2 cancers-17-02226-f002:**
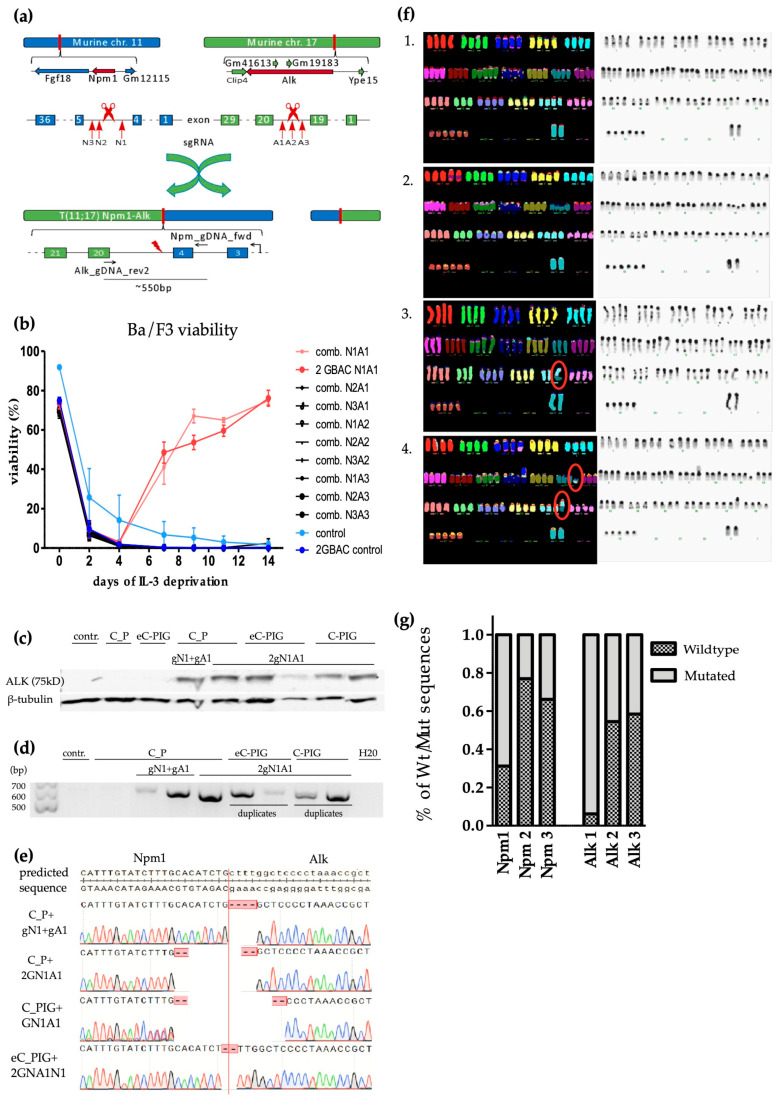
(**a**) Schematic representation of *Npm-Alk* translocation in murine cells: The sgRNAs (red arrows) target specific introns within the *Npm1* and *Alk* locus. By inducing double-stranded breaks the murine chromosomes 11 and 17 can translocate during repair. The breakpoint of *Npm1-Alk* is located in the 4th exon of *Npm1* and the 20th exon of *Alk*. A PCR covering the region around the breakpoint was designed using a *Npm1*-specific forward primer (NPM_gDNA_fwd) and an *Alk*-specific reverse primer (ALK_gDNA_rev2), indicated in the graph as black arrows. A1, A2, A3: sgRNAs targeting *Alk*; N1, N2, N3: sgRNAs targeting *Npm*1. (**b**) Flow cytometric monitoring of cell viability of IL-3 deprived Ba/F3 cells after transduction with Cas9 and sgRNA retroviral vectors either from 2 separate plasmids expressing a single gRNA (comb) or the dual gRNA expression vector 2GBAC. All 9 possible combinations of 3 gRNAs targeting *Npm*1 and 3 gRNAs targeting *Alk* were tested. (**c**) Alk Western blot of Ba/F3 cell lysates from cell lines expressing different constitutive Cas9 constructs with or without the indicated sgRNAs. Β-tubulin expression was used as control. (C_P: MSCV_Cas9_Puro; C_PIG: Cas9_PIG; eC_PIG: eCas9_PIG; gA1: GBACy_Alk_1; gN1: GBAC_Npm1_1, 2GN1A1: 2GBAC_Npm1-Alk). (**d**) PCR covering the translocation site with a forward primer in the *Npm1* intron 250 bp upstream of the gRNA cutting site and a reverse primer in the Alk intron 250 bp downstream of the Alk gRNA site. Genomic DNA from Ba/F3 cells transduced with the indicated constructs was used as template for the PCR reaction. (C_P: MSCV_Cas9_Puro; C_PIG: MSCV_Cas9_PIG; eC_PIG: MSCV_eCas9_PIG; gA1: GBACy_Alk_1; gN1: GBAC_Npm1_1, 2GN1A1: 2GBAC_Npm1_1+Alk_1). (**e**) Sanger-Sequencing of the selected PCR products from (**d**). Sequences are aligned to the predicted sequence produced by direct fusion of *Npm1* and *Alk* at the respective gRNA cutting site. (**f**) M-FISH analysis of viable Ba/F3 cells. Karyotyping of the genome of Ba/F3 cells with no infection (1), overexpressing plasmid *Npm1-Alk* (2), infected with a Cas9 vector and double infected with sgRNA vectors (3) and infected with a Cas9 vector and a single infection of dual sgRNA vector (4). Shown are one out of four assays performed each. Ba/F3 cells double infected with sgRNA vectors (3) and Ba/F3 cells infected with a dual sgRNA vector (4) presented with a t(11;17) chromosomal translocation (marked with a red circle). Uninfected Ba/F3 cells (1) and Ba/F3 cells overexpressing *Npm1-Alk* (2) were used as control cell lines. (**g**) Cutpoint efficiency analysis: Schematic ratio of mutated gDNA to wildtype gDNA of cells infected with either GBAC Npm1_1 (Npm1), Npm1_2 (Npm2), Npm1_3 (Npm3), Alk_1 (Alk1), Alk_2 (Alk2) or Alk_3 (Alk3). Original western blots are presented in Appendix A.

**Figure 3 cancers-17-02226-f003:**
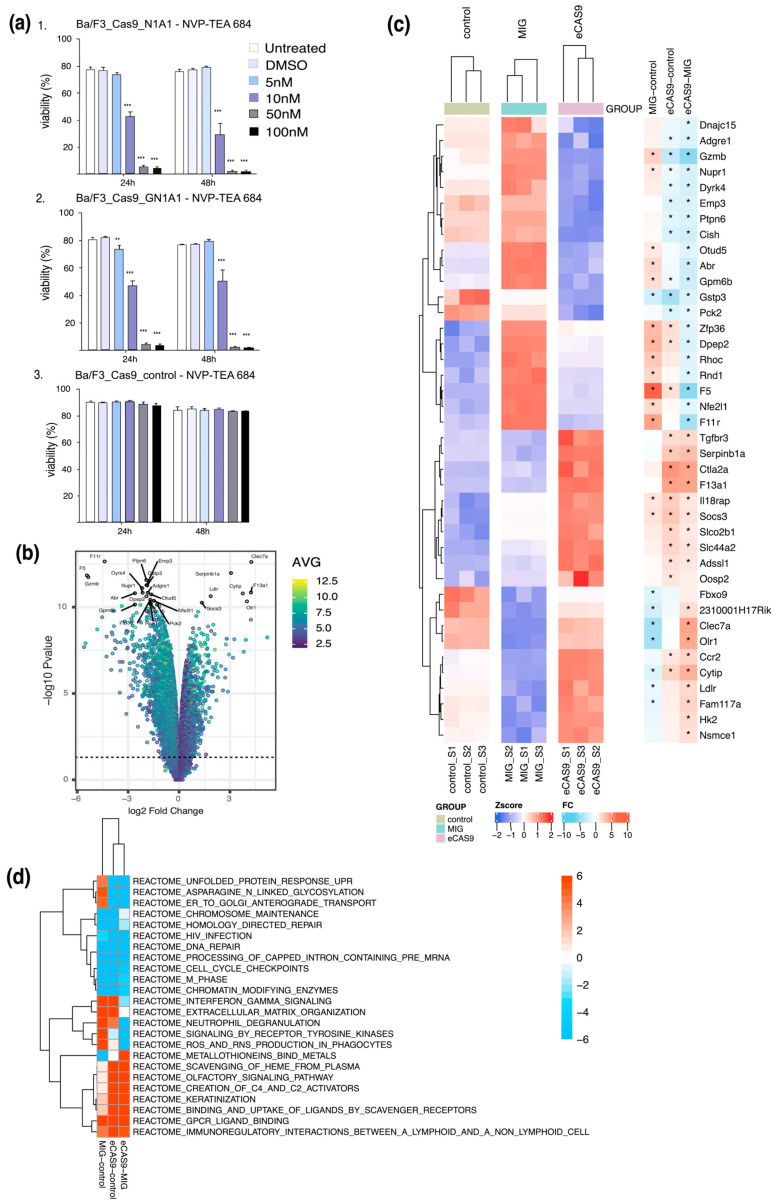
(**a**) Viability assay of Ba/F3_Cas9_N1A1 (1.) and Ba/F3_Cas9_2GN1A1 (2.) cells treated with the indicated concentrations of the Alk inhibitor (NVP-TAE648). Viability was monitored via flow cytometry after 24 and 48 h (data in technical triplicates). Ba/F3_Cas9 cells grown in the presence of IL-3 were used as control (3.). Statistical significance of the viability differences *Npm-Alk* rearranged cells at each concentration compared to the untreated control are highlighted *p* < 0.01 (**) or *p* < 0.001 (***). (**b**) Volcanoe plot of differentially regulated genes in Ba/F3 control, Ba/F3 MIG-NA and Ba/F3 Cas9-2GN1A1 cells determined by microarray gene expression analysis. The graph shows log2 fold up- or downregulation plotted against the FDR-adjusted *p*-value (−log10) in Ba/F3 Cas9-2GN1A1 cells compared to Ba/F3-MIG-NA cells. The horizontal dashed line indicates the statistical significance threshold at −log10(p) = 1.3 (corresponding to *p* = 0.05). (**c**) Heatmap of gene expression changes showing the top 40 differentially expressed genes between *Npm-Alk* overexpressing cells (MIG) and *Npm-Alk* rearranged cells (eCAS9). (**d**) Gene set enrichment analysis (GSEA) of differentially expressed genes between MIG and eCAS9 cells based on the REACTOME [38] pathway database.

## Data Availability

The data presented in this study are available on request from the corresponding author.

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
