# Peer review of "Modeling the t(2;5) Translocation of Anaplastic Large Cell Lymphoma Using CRISPR-Mediated Chromosomal Engineering"

_cancers, 2025, doi:10.3390/cancers17132226_

Round 1
Reviewer 1 Report
Comments and Suggestions for Authors
To the authors,
Khan, et al present interesting research in which they created t(2;5) chromosomal translocation by CRISPR-Cas9 system. They have improved their strategy to create this fusion gene model more efficiently. Creating NPM::ALK by CRISPR-Cas9 might be novel, but since genome editing systems became available to induce double strand break, there have been several reports on generating chromosomal translocation with those systems, such as zinc finger, TALEN, or CRISPR-Cas. At least they should mention earlier reports and explain how their strategy is different than previous reports (or similar). The Introduction and Discussion sections are generally lacking in depth. Addressing these concerns is necessary for the manuscript to be considered for publication in this journal. Otherwise, the authors should focus on the methodological aspects and consider submitting to a journal that specializes in methodology.
Major points:
- Despite there are several papers which are about creating chromosomal translocation with gene editing systems, the authors seem not to pay attention to those earlier papers, such as Tamai, et al Cancer Gene Ther 2023; Torres, et al. Nat Commun 2014; Choi, et al. Nat Commun 2014, They should mention these earlier reports in their introduction or discussion.
- Figure 3 was not discussed enough in the context. The manuscript fails to address these data presented, merely stating that there are differences in gene expression between the two cell types without providing any detail on the nature of these differences.
- Discussion should not be just summary of the results.
Minor points:
- Figure should be in the order to appear (in the text fig 1b came earlier than fig1a). There is no Fig 1c in the context.
- Line 298: Please correct “able survive” to “able to survive”
- Fig2b needs to show wild type ALK (200kDA size) together
- Fig 3a,b: Please add the control result to the figure.
- Fig3c legend is missing
Author Response
We would like to thank you for the careful evaluation of our work and for the constructive comments and suggestions, which have helped us to improve the quality and clarity of our manuscript.
In the revised version, we have now addressed your comments and made significant enhancements to both the Results and Discussion sections. We have substantially revised Figure 3, which now also includes a GSEA analysis of the gene expression differences in NPM-ALK translocated cells compared to NPM-ALK overexpressing cells. Furthermore, the supplementary figures have been added and effectively support the main findings of the manuscript.
Our detailed responses to the key comments are listed below.
Point-by-Point response:
- Despite there are several papers which are about creating chromosomal translocation with gene editing systems, the authors seem not to pay attention to those earlier papers, such as Tamai, et al Cancer Gene Ther 2023; Torres, et al. Nat Commun 2014; Choi, et al. Nat Commun 2014, They should mention these earlier reports in their introduction or discussion.
Response:
We apologize for any unintended omissions in the reference list. We have now updated the references and included the suggested publications in the manuscript´s introduction. The included text section reads as follows: “ In recent years, several studies have demonstrated the use of gene editing systems, such as transcription activator-like effector nucleases (TALEN) and CRISPR-Cas9, to induce specific chromosomal translocations and intrachromosomal inversions in vitro, including MLL-AF9 (Schneidawind et al., Blood Advances 2018), BCR-ABL (Tamai et al., Cancer Gene Ther 2023), RUNX1-RUNX1T1 (Torres et al., Nat Commun 2014) and others (Choi et al., Nat Commun 2014) (Sarrou et al., Int J Mol Sci 2020)”.
- Figure 3 was not discussed enough in the context. The manuscript fails to address these data presented, merely stating that there are differences in gene expression between the two cell types without providing any detail on the nature of these differences.
Response:
Through an inadvertent error, a previous version of Figure 3 was included in the first submission. An updated version including a Gene Set Analysis (GSEA) of the gene expression array data (now Fig. 3d) was added to Figure 3, the former Fig. 3e was moved to the supplementary data (supp. Fig 8). Furthermore, an analysis of the chromosome-specific gene expression changes was included in the supplementary data.
- Figure should be in the order to appear (in the text fig 1b came earlier than fig1a). There is no Fig 1c in the context.
Response:
We have revised the figure order so that the figures now appear in the sequence in which they are first mentioned in the text. Figure 1c has been moved and is now presented as Figure 2a in the revised manuscript.
- Fig2b needs to show wild type ALK (200kDA size) together
Response:
All the uncropped Western Blots, including the uncropped ALK Blot have been added to the supplementary data section. (Suppl. Figure 10)
- Fig 3a.b: Please add the control result to the figure.
Response:
The control results have now been added to Figure 3a and are specifically presented under the new subpanel labeled Figure 3a3
- Discussion should not be just summary of the results.
Response:
Both the results and discussion sections have been thoroughly revised, with an expanded emphasis on the GSEA analysis of the gene expression differences in NPM-ALK translocated cells compared to NPM-ALK overexpressing cells to improve clarity and avoid replication.
- Figure legend 3c is missing
Response: The figure legend has been added and can now be found under Fig. 3d. The legend reads: Volcanoe plot of differentially regulated genes in Ba/F3 control, Ba/F3 MIG-NA and Ba/F3 Cas9-2GN1A1 cells determined by microarray gene expression analysis. The graph shows log2 fold up- or downregulation plotted against the FDR-adjusted p-value (-log10) in Ba/F3 Cas9-2GN1A1 cells compared to Ba/F3-MIG-NA cells.
All figure references and legends have been updated accordingly to ensure clarity and consistency in the manuscript
We believe that these revisions have significantly strengthened our manuscript and hope that it is now suitable for publication in Cancers.
Reviewer 2 Report
Comments and Suggestions for Authors
The manuscript aims to simulate the T (2;5) chromosomal rearrangement of ALK+ anaplastic large cell lymphoma (ALCL) in mouse cell lines using CRISPR/Cas9 technology, and to study the mechanism of T (2;5) chromosomal rearrangement of ALK+ anaplastic large cell lymphoma, to more accurately reproduce the expression of the NPM-ALK fusion gene in this disease and its oncogenic mechanism. The authors successfully developed a CRISPR/Cas9-based system to induce the Npm-Alk fusion gene, providing evidence of its oncogenic potential. They also confirmed the functions of the fusion gene. Importantly, the CRISPR-mediated Npm-Alk translocation models exhibited distinct gene expression patterns compared to conventional Npm-Alk cDNA overexpression models. This suggests that CRISPR-mediated Npm-Alk translocation holds promise for gene expression analysis. Notably, changes in gene expression near the chromosomal translocation point indicate that chromosomal structural alterations significantly impact gene expression. The manuscript is well-written and clearly presented, making it suitable for publication in Cancers with minor revisions. Before publication, the following points should be addressed:
- In lines 232-238: Despite observed baseline Cas9 expression, the authors should compare the expression levels between the constitutive and tet-inducible systems and explain why they did not optimize the tet-inducible system to control Cas9 expression?
- Lines 284-287 state that the original Cas9 construct showed increased toxicity in certain cells (data not shown). The authors should provide a plausible explanation for this observation?
- In Chapter 3.3, while the authors demonstrated the efficiency of the sgRNA combination targeting Npm and Alk, only one specific combination successfully induced Npm-Alk chromosomal translocations. The authors should explore the mechanism behind this result?
Author Response
We would like to thank you for the careful evaluation of our work and for the constructive comments and suggestions, which have helped us to improve the quality and clarity of our manuscript.
In the revised version, we have now addressed your comments and made significant enhancements to both the Results and Discussion sections. We have substantially revised Figure 3, which now also includes a GSEA analysis of the gene expression differences in NPM-ALK translocated cells compared to NPM-ALK overexpressing cells. Furthermore, the supplementary figures have been added and effectively support the main findings of the manuscript.
Our detailed responses to the key comments are listed below.
We believe that these revisions have significantly strengthened our manuscript and hope that it is now suitable for publication in Cancers.
Point-by-Point response:
- In lines 232-238: Despite observed baseline Cas9 expression, the authors should compare the expression levels between the constitutive and tet-inducible systems and explain why they did not optimize the tet-inducible system to control Cas9 expression?
Response:
The following section was added to the manuscript:
In our initial analysis of the different inducible and constitutive Cas9 expression systems we had designed, we observed a significant leakiness of the tet-system, irrespective of the Cas9 variant (Cas9 or eCas9), in spite of our using a third-generation tet-on system (Loew et al., BMC Biotechnology 2010 Nov 24:10:81). This is shown e.g. in Fig. 1c, where residual expression can be detected in K562 and Ba/F3 cells. Of note, we did not observe any expression in the primary murine lymphoma cell line Mcl 22-1, due to toxic effects of Cas9 expression in this cell line. Consistently, the constitutive Cas9 expression system showed a higher efficiency, manifested by higher depletion rates when targeting essential genes such as Rpa3, likely due to counterselection and outgrowth of poor inducer cells in the tet-inducible system (data not shown). We therefore decided to proceed using the constitutive Cas9 expression variants for further testing, since the combination of leakiness and toxicity limited the advantage of the tet-inducible system over the constitutive variants.
A comparison is featured in the supplementary data (supp. Fig. 1+2)
- Lines 284-287 state that the original Cas9 construct showed increased toxicity in certain cells (data not shown). The authors should provide a plausible explanation for this observation?
Response:
The following section was added to the manuscript:
The increased toxicity observed with the original Cas9 construct in certain cells has been described by other groups as well, and seems to be associated with its activation of a p53-mediated DNA damage response (DDR) (Haapaniemi et al. Nature Medicine 2018, Alvarez et al., Nat Commun. 2022, Morgens et al., Nature Commun. 2017). When CRISPR-Cas9 induces double-strand breaks (DSBs), cells with functional p53 pathways respond by initiating cell cycle arrest or apoptosis to mitigate genomic instability. Accordingly, we observed significant toxicity of Cas9 expression in primary cells (murine BM cells) or primary cell lines (e.g. the primary murine lymphoma line Mcl 22-1), which may retain more elements of their DDR including the p53 pathway compared to established cell lines.
- In Chapter 3.3, while the authors demonstrated the efficiency of the sgRNA combination targeting Npm and Alk, only one specific combination successfully induced Npm-Alk chromosomal translocations. The authors should explore the mechanism behind this result?
Response:
To analyze the differences in efficacy between the different gRNAs used to target the NPM and ALK locus, we performed a semiquantitative cutpoint-analysis by deep-sequencing. The analysis showed a profound difference in cutting efficiency, with the best gRNAs (gN1 and gA1) reaching a cut rate of 70% and 90%. We have included this analysis in the results section and also discuss the implications in the 4th paragraph of the discussion.
We believe that these revisions have significantly strengthened our manuscript and hope that it is now suitable for publication in Cancers.
Reviewer 3 Report
Comments and Suggestions for Authors
In their manuscript, Khan R and coauthors describe a new model for anaplastic large cell lymphoma with more accurate modeling of the Npm1-Alk translocation. The author has nicely demonstrated that the model reproduces the key features of this type of lymphoma such as ALK inhibitor dependency and IL-3 independence. Despite the interesting data obtained, there are some comments and errors to be addressed:
- The manuscript would benefit from the introduction of a comparison of the gN1-gA1 generated translocation with the schematics (sequence) of the most frequent Npm1-Alk translocations in patients.
- The Ba/F3 cell line should be described in more detail, as it is critical for understanding the limitations of this model and for analyzing differences in the transcriptome.
- The results with HL60 and K562 (Fig1d) should be explained in the text.
- The experiment targeting Rpa3 with gRNA is irrelevant to the main story. Please remove this part to the Supplementary section.
- There are errors in Fig. 3. Fig. 3c is not described in the legend.
- The supplementary file is unreadable. It is not in final form. Some figures are in the "Review" panel on the right.
Author Response
We would like to thank you for the careful evaluation of our work and for the constructive comments and suggestions, which have helped us to improve the quality and clarity of our manuscript.
In the revised version, we have now addressed your comments and made significant enhancements to both the Results and Discussion sections. We have substantially revised Figure 3, which now also includes a GSEA analysis of the gene expression differences in NPM-ALK translocated cells compared to NPM-ALK overexpressing cells. Furthermore, the supplementary figures have been added and effectively support the main findings of the manuscript.
Our detailed responses to the key comments are listed below.
Point-by-Point response:
- The manuscript would benefit from the introduction of a comparison of the gN1-gA1 generated translocation with the schematics (sequence) of the most frequent Npm1-Alk translocations in patients.
Response:
Following the reviewer’s suggestion, we have compared the breakpoints and fusion sequence in our model with the canonical human NPM1-ALK translocation. The translocation t(2;5)(p23;q35) fuses exon 4 of NPM1 to exon 20 of ALK, with the breakpoint of NPM1 predominantly located within intron 4 encompassing a region of approximately 1 kilobase, and the breakpoint of ALK typically occurring within intron 19 spanning about 2,2 kilobases (Krumbholz et al., Oncotarget 2018). In our CRISPR-induced murine model, most breakpoints occurred in close proximity to the gRNA cutting site as expected. A schematic overview was added to the supplementary figures (suppl Fig. 6).
- The Ba/F3 cell line should be described in more detail, as it is critical for understanding the limitations of this model and for analyzing differences in the transcriptome.
Response: A more detailed description of Ba/F3 cells has been added to the Materials and Methods section under "Culturing of cell lines," as well as in the first paragraph of Section 3.2 in the Results.
- The results with HL60 and K562 (Fig1d) should be explained in the text.
Response:
We have added a section for K562 and HL-60 cells to the Materials and Methods section under "Culturing of cell lines," as well as to the second paragraph of Section 3.2 in the Results. The revised text reads as follows:
During testing of the tet-inducible Cas9 expression constructs, Ba/F3 and K562 cells showed strong Cas9 expression levels under doxycycline (dox) treatment (+), with low but residual Cas9 expression levels in the absence of doxycycline (-) (Fig. 1c, upper panel). Conversely, no relevant Cas9 expression could be shown in HL-60 and MCL 22-1 samples irrespective of dox treatment
- The experiment targeting Rpa3 with gRNA is irrelevant to the main story. Please remove this part to the Supplementary section.
Response:
The experiment targeting the essential gene RPA3 was included solely as a technical control to validate the functional integrity of the system, and was not intended for hypothesis testing.
We believe that these revisions have significantly strengthened our manuscript and hope that it is now suitable for publication in Cancers.
Round 2
Reviewer 1 Report
Comments and Suggestions for Authors
The authors have addressed all of my concerns. I have no further requests.
Reviewer 3 Report
Comments and Suggestions for Authors
The manuscript was significantly improved after revision. The reviewer's comments have been addressed.